# VHCF, Tribology Characteristics and UNSM Effects of Bainite and Martensite Spring Steels

**Min Soo Suh** [1], **Seung Hoon Nahm** [2], **Chang Min Suh** [3,*] and **Young Sik Pyun** [4,*]

1    Korea Institute of Energy Research, Daejeon 34129, Korea; mssuh@kier.re.kr
2    Korea Research Institute of Standards and Science, Daejeon 34113, Korea; shnahm@kriss.re.kr
3    School of Mechanical Engineering, Kyungpook National University, Daegu 41566, Korea
4    Department of Fusion Science and Technology, Sun Moon University, Asan 31460, Korea
*    Correspondence: cmsuh@knu.ac.kr (C.M.S.); pyoun@sunmoon.ac.kr (Y.S.P.)

**Abstract:** It has been reported that the duplex bainite microstructure obtained by austempering (AT) shows higher strength, ductility and impact toughness than quench and tempered (QT) martensite structure in SAE9254 spring steel. However, there seems to be no research on the very high cycle fatigue (VHCF) and tribology characteristics of bainite structure for durability design of next generation spring steel from the perspective of engineering and industrial applications. This is a follow-up study that quantitatively analyzed the mechanical properties, microstructural deformation characteristics, and impact toughness of bainite and martensite using EBSD (Electron Backscatter Diffraction) and SEM (Scanning Electron Microscope) analyses. In this study, VHCF, HCF, tribology characteristics and UNSM (ultrasonic nanocrystal surface modification) effects under duplex bainite and single martensite microstructures were quantitatively studied and analyzed by fracture mechanics from the engineering and industrial point of view to improve durability and weight reduction in spring steels. The bainite AT and martensite QT specimens showed a 56% and 33% increase in fatigue limit for as received AR specimens. Fisheye cracks in duplex bainite AT specimens are similar to 'facet internal cracks' that initiated in the absence of inclusions. Generally fisheye crack fracture mode is preferred in VHCF, but fisheye crack was not found in the QT and the AR specimens at all. The UNSM-treated specimens showed fatigue limits that were about 33~50% higher than the untreated specimens.

**Keywords:** spring steel; bainite; martensite; austempering; very high cycle fatigue; tribology; ultrasonic nanocrystal surface modification



## 1. Introduction

Recent studies about duplex microstructures and mechanical properties of spring steels after quenching and tempering (QT) and austempering (AT) reported that the poly phase structures with bainite, martensite and retained austenite showed high strength, ductility and impact toughness [1–14]. The final microstructure of this spring steel is tempered martensite by QT heat treatment, but this microstructure showed the basic problem of hydrogen embrittlement when strengthening. Therefore, if the final tempered martensite microstructure of spring steel can be replaced by duplex bainite microstructure, it will be very helpful in the development of high strength and long-life products.

Nie et al. [1] reported that mixed microstructure of bainite and martensite showed much superior fatigue strength to an only martensite microstructure in the case of spring steels.

Suh et al. [2] studied the impact toughness of spring steels after transformation of bainite and martensite and reported that the duplex bainite microstructure exhibits better mechanical properties and impact behavior than the single martensite microstructure.

A study about fatigue behavior of high strength steel with duplex phase of bainite and martensite by Wei et al. [13] reported that it showed higher fatigue strength and lower crack propagation rate. These findings appear to be very useful in practice and applied engineering.

However, it seems that there are few systematic studies on the Very High Cycle Fatigue (VHCF) and friction properties of duplex bainite microstructured steel required for durability design of next-generation spring steel [15–28]. In particular, there seems to be no study on the surface treatment effects of UNSM (Ultrasonic Nanocrystal Surface Modification) on duplex bainite microstructured steel [29–33].

Therefore, in this study, considering the practical viewpoints for the duplex bainite microstructure and the single martensite structure, it is judged whether or not it can be effectively applied to the industry from the structural and engineering viewpoints.

This is a follow-up study that quantitatively analyzed the mechanical properties, microstructural deformation characteristics, and impact toughness of duplex bainite and single martensite. In the previous study, these factors were quantitatively analyzed using EBSD and SEM analyses, and it was confirmed that they were duplex bainite and single martensite microstructures [2].

(a) The change in impact toughness and mechanical properties according to the duplex bainite and single martensite formation of spring steel, which is important for engineering and industrial applications but has few published data; (b) quantitative analysis of lath length and width after duplex bainite and single martensite formation; (c) quantitative analysis of phase transformation properties; and, (d) comparative analysis of duplex bainite and single martensite formation with a pole figure.

In this study, the VHCF, HCF, tribological properties, and UNSM effects of spring steels with bainitic and martensitic microstructures were quantitatively studied through fracture mechanics and fracture surface analysis methods from the engineering and industrial point of view to improve durability and weight reduction in spring steels of SAE9254 confirmed in the previous study.

Therefore, the following studies were conducted according to microstructures in this study. In other words, (a) Comparison of S-N curves and VHCF characteristics; (b) Comparison of the UNSM effects according to microstructures; (c) SEM observation and analysis of fatigue fractured surfaces; (d) Variation in friction coefficients, wear amounts, specific wear rates and SEM analysis of wear surface; and (e) The excellent UNSM effects were verified on the VHCF and wear properties of spring steels with bainitic and martensitic microstructures.

## 2. Materials and Experimental Methods

### 2.1. Test Materials and Heat Treatment Method

The spring steel used in this study was SAE9254 (similar to SUP12), and the chemical composition of the AR spring steel was 0.55% C, 1.5% Si, 0.7% Mn, and 0.7% Cr. The received specimen was called AR (as received), and QT (quenched and tempered) and AT (austempered) specimens were made with heat treatment cycles.

Round AR spring steel was cut to a diameter of 13 mm and a length of 110 mm. The QT heat treatment used in this study was austenitized in the air at 980 °C for 15 min, water-quenched at 60 °C for 5 min, salt bathed at 430 °C for 90 min, and then air-cooled. Subsequently, the same specimen was austenitized at 980 °C for 15 min, salt bathed at 300 °C for 30 min, and then air-cooled for austenitization of AT specimen. QT martensite specimens and AT bainite specimens were obtained by salt bathing.

Since the Ms temperature of the specimen was evaluated as $265 \pm 4$ °C and the AC line temperature as $755 \pm 5$ °C, martensite transformation did not occur during austempering. These heat treatment cycles were suggested by the manufacturer. Fatigue, tensile and tribology specimens were prepared using the heat-treated specimens by precision machining.

### 2.2. Experimental Methods

The spring steel was machined into a tensile test specimen (ASTM A370). Tensile tests were performed using a universal testing machine (Autograph AGS-X, Shimadzu, Kyoto, Japan) at a strain rate of 0.02 mm/s according to the specifications of ASTM A370.

The hardness of the specimen was measured using a Vickers hardness tester (JP/HM-112, Mitutoyo, Kawasaki, Japan) and measured 12 times under a 5 kN load, primary and secondary, based on the center. In addition, to observe the microstructure and measure the hardness, we selected a part of the test specimen that was not stressed or deformed. An etching solution of 5 mL HCl, 1 g picric acid, and 100 mL ethanol (95%) was used to observe the optical microstructure (OM).

EBSD of the sample was performed after electropolishing with a mixed solution of 10% perchloric acid and 90% acetic acid. No additional cleaning processes were required. In this study, an EBSD measurement system (DigiView EBSD Camera, EDAX, Mahwah, NJ, USA) was used, and measurement conditions with an accelerating voltage of 18 kV were used. The crystal size was set to 0.3 μm or larger.

The diameter and radius of curvature of the fatigue specimen are 3.0 mm and 7.0 mm, respectively. In addition, the stress concentration factor Kt of the fatigue specimen is 1.02. Specimens were polished using # 100 to # 2000 abrasive paper to obtain mirror surfaces according to the purpose.

Fatigue and VHCF characteristics at room temperature were obtained using a cantilever rotating bending fatigue tester (YRB200, Yamamoto, Nagaoka, Japan) as shown in Figure 1A. This tester can test four specimens simultaneously under 53 Hz and stress ratio R = −1 in air. The fractured fatigue specimens were magnified and analyzed by fracture surface analysis using a SEM (S-4200, Hitachi, Tokyo, Japan) and EDS (energy dispersive spectroscopy, Horiba, Kyoto, Japan) analysis equipment.

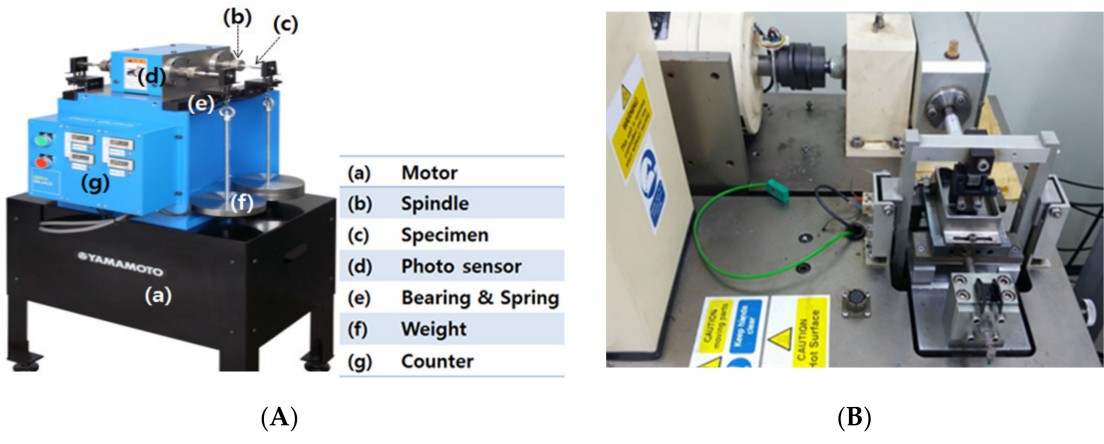

(**A**)             (**B**)

**Figure 1.** Rotating fatigue testing machine (**A**) and reciprocating tribology test machine (**B**).

Since the UNSM treatment technology [29–33] used in this study hits the metal surface more than 20,000 times per second, it becomes a nanostructured metal surface while forming fine dimples, and the surface roughness is improved while compressive residual stress is formed deeply. Table 1 shows the UNSM treatment conditions applied to this study.

**Table 1.** UNSM conditions.

| Specimen | Frequency, kHz | Generator Power Level, % | Static Level, N | Feed Rate, mm/rev | Speed, rpm | Tip Diameter, mm |
|---|---|---|---|---|---|---|
| AR | 20 | 30 | 15 | 0.04 | 120 | 2.38 |
| AT | 20 | 30 | 30 | 0.04 | 120 | 2.38 |
| QT | 20 | 30 | 30 | 0.04 | 120 | 2.38 |

Rectangular wear specimens of 13 W × 58 L × 4 T (mm) were machined from a bar of 13 mm diameter for this study. Tribology test conditions are shown in Table 2, and the reciprocating tribology test machine (TE77 AUTO, Plint & Partners, Newbury, UK) used for this study is shown in Figure 1B. Tribology test was conducted at dry condition by

applying 50 N and 100 N for 1800 s and 1200 s, and tribology characteristics were compared depending on the duplex bainite and single martensite spring steels of the rectangular wear specimens.

**Table 2.** Tribology test conditions.

| Load, N | Frequency, Hz | Stroke, mm | Time, s | Counter Part, SAE 52,100 | Condition |
|---|---|---|---|---|---|
| 50, 100 | | | 1800 | | Untreated, dry |
| 100 | 2 | 15 | 1200 | 10 mm | UNSM treated, dry |

## 3. Test Results and Discussion

### 3.1. Tensile Test

The tensile test results are summarized in Table 3. The AR specimen obtained a maximum tensile strength of 991.5 MPa, a yield strength of 526.5 MPa, and an elongation rate of 19.5% in the tensile test. Tensile strengths of the AT and QT specimens are measured at 1824.4 MPa and 1723.1 MPa, 84% and 73.8% higher than the AR specimens.

**Table 3.** Mechanical properties.

| Specimen | Tensile Strength, MPa | Yield Strength, 0.2%, MPa | Elongation, % | Reduction in Area, % |
|---|---|---|---|---|
| AR | 992 | 527 | 20 | 52 |
| AT | 1824 | 1682 | 13 | 50 |
| QT | 1723 | 1612 | 12 | 40 |

In addition, the duplex bainite AT specimen showed a 5.9% higher tensile strength than the single martensite QT specimen. The increase in tensile strength of AT and QT compared to the AR specimen resulted from the change in microstructure owing to the formation of duplex bainite and single martensite by the heat treatment.

The elongation and area reduction rates predominate despite the increase in tensile strength. Usually, as the tensile strength increases, the elongation decreases and the area reduction rate also decreases, but the duplex bainite AT specimen showed increased properties compared to the single martensite QT. Such physical properties seem to be greatly influenced by the size and shape of the lath [2].

### 3.2. Hardness Test

Table 4 summarizes the average Vickers hardness values of the specimens used in this study. The AR specimen showed a standard deviation (S.D.) of 12.7, the AT specimen showed the lowest values of 10.1 and the QT specimen showed 14.9. The bainite AT and martensite QT specimens showed an increase in hardness of 102.6% and 84.5%, respectively, compared to the AR specimen.

**Table 4.** Results of Vickers hardness test.

| Cond. | AR | AT | QT |
|---|---|---|---|
| Hv | 276 | 559 | 509 |
| S.D. | 12.7 | 10.1 | 14.9 |

In addition, the AT specimen showed a 9.8% increase compared to the QT specimen. The significant increase in AT and QT hardness values compared to that of the AR specimen resulted from the duplex microstructure of bainite and single martensite phase structures and the microscopic transformation according to the heat treatment condition.

### 3.3. Microstructural Transformation

3.3.1. Variation in Microstructure by the Heat Treatment Cycles

Figure 2a shows an optical microscope (OM) photograph (500×) of the AR specimen, showing a mixed structure of ferrite (white color), pearlite (gray color), and cementite with an average grain diameter of approximately 11 μm.

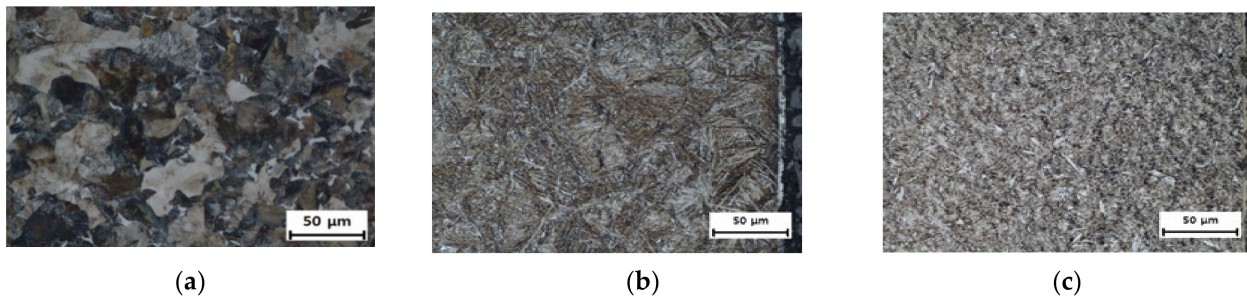

(**a**)                                                        (**b**)                                                        (**c**)

**Figure 2.** OM (optical microscope) photographs of (**a**) AR, (**b**) AT, and (**c**) QT specimens.

Figure 2b shows the OM structure (500×) of the AT specimen after the heat treatment cycle. In the AT specimen, the needle shaped laths were developed well within a prior austenite grain boundary (PAGB) owing to the microstructural formation, with bainite predominantly formed.

Figure 2c shows the OM structure (500×) of the QT specimen, which shows a refined microstructure after the heat treatment cycle. Some ferrites were formed owing to compositional heterogeneity, and the rest were transformed into martensite, resulting in white areas appearing in the QT specimen.

3.3.2. Electron Microscopy Observation According to Microstructure

Figure 3 shows the microstructures observed using a scanning electron microscope. The bainite and some retained austenite (see arrows) structures were observed at 20,000× in Figure 3a of the AT specimen. Some other studies reported that duplex phase structures of bainite and martensite were also obtained by austempering [1–14].

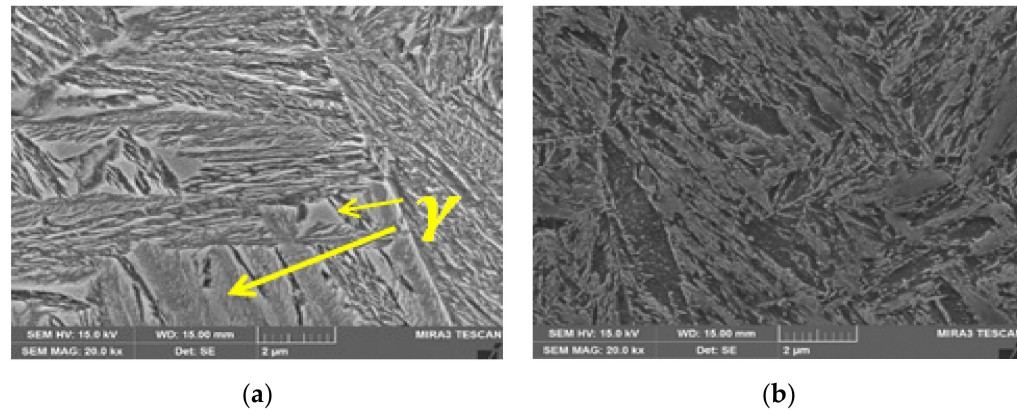

(**a**)                                                        (**b**)

**Figure 3.** SEM photographs at 20,000× of (**a**) AT and (**b**) QT specimens.

Figure 3b is a photograph of the QT specimen at 1.5 mm from the surface, where martensite and tempered martensite structures is well observed at 20,000×.

### 3.4. EBSD Analysis According to Microstructure

3.4.1. EBSD Analysis of the AR Specimen

Figure 4 shows the AR specimen data using EBSD, and the average grain size of the microstructure is approximately 11 μm. Figure 4 shows a phase map showing the crystal

orientation with isotropic grains, and shows two phases of 76.9% ferrite (red color) and 23% $Fe_3C$ (green color). This figure shows that the microstructures of the AR specimen of the spring steel used in this study have a uniform orientation.

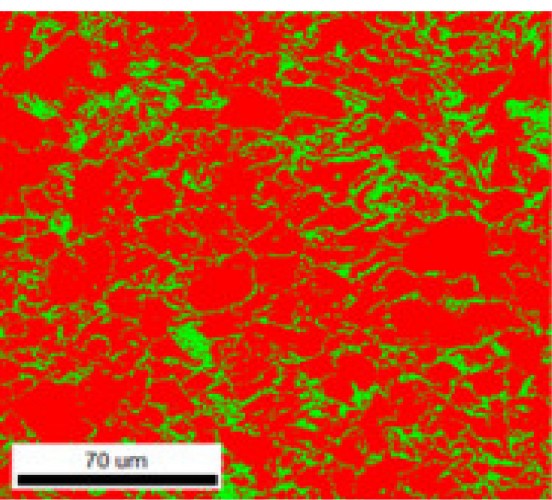

**Figure 4.** Phase map of EBSD of the AR specimen (Scale bar is 70 μm).

### 3.4.2. EBSD Analysis of the AT Specimen

Figure 5 shows the inverse pole figure map and phase map of the AT specimen examined with EBSD. Some locally dark areas were observed among many laths, which were removed by filtering. The length of the long plate type lath was measured at a maximum of 29.4 μm, a minimum of 7.1 μm, and an average of 15.5 μm due to its non-uniform chemical composition. The width of lath was measured at a maximum of 5.29 μm, a minimum of 0.59 μm, and an average of 1.76 μm. In addition, the AT specimen showed about 6.5% of the FCC γ-Fe ratio, which is much larger than 1% of the QT specimen. However, the fraction of BCC α-Fe is 93.5%, which shows that existing austenite was not fully transformed to bainite and a considerable amount of retained austenite occurred, as seen in Figure 3a.

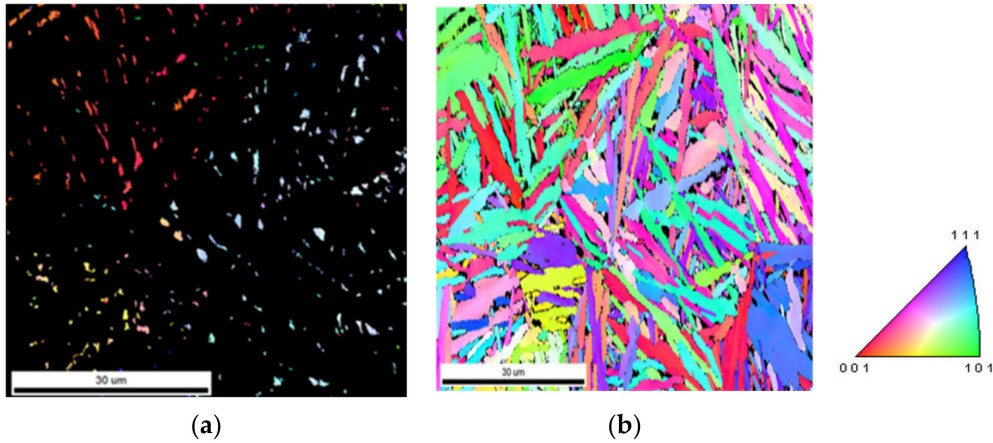

(**a**)  (**b**)

**Figure 5.** Inverse pole figure map (**a**) and phase map (**b**) of austenite and ferrite of the AT specimen. (Scale bar is 30 μm).

Figure 5a shows the IPF maps of FCC (face centered cubic) and BCC (body centered cubic). Figure 5b illustrates the shape and orientation of bainite made of lath in various cases. About 6.5% of this FCC phase, γ-Fe, is present. The inside of the prior austenite grain region has a locally similar crystal orientation. It can be seen that the laths have similar

colors locally, but it can also be confirmed that they do not only appear in the same color. In addition, a phenomenon with a slightly different internal crystal orientation can also be confirmed. These factors lead to consideration of two degrees of possibilities.

First, various austenite crystals are generated in the austenitic process, and phase transformation occurs in the cooling process. Different orientations are clearly represented by different colors.

Second, the different orientation is due to the behavior inside the same prior austenite crystal during phase transformation. When bainite is generated by the phase transformation phenomenon, microplastic deformation is applied to the surrounding austenite. This is because such microplastic deformation applies a deformation of the local crystal orientation of austenite. This enables a more accurate analysis by looking at and judging the orientation of retained austenite together with the method of tracing prior austenite grain boundaries (PAGB) with the Kurdjumov–Sachs relationship orientation [2].

### 3.4.3. EBSD Analysis of the QT Specimen

Figure 6 shows the inverse pole figure map and phase map of the QT specimen, where red color is α-Fe with BCC crystal structure, light green color is γ-Fe with FCC crystal structure, and black color is the area not expressed because of inappropriate filtering condition. A lath martensite phase with a BCT (body centered tetragonal) structure with an average width of less than 0.81 μm (maximum 2.76 μm, minimum 0.55 μm) was identified. EBSD equipment used in this study recognizes all BCT phases as BCC crystal structure.

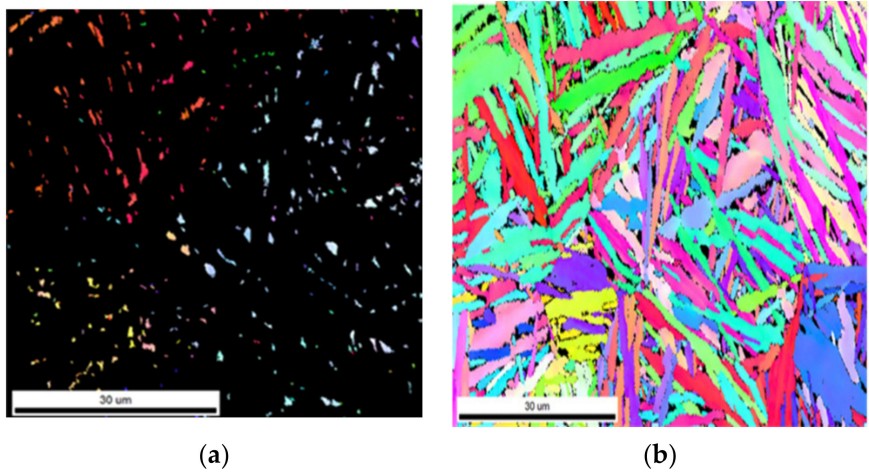

(**a**)    (**b**)

**Figure 6.** Inverse pole figure map (**a**) and phase map (**b**) of austenite and ferrite of the QT specimen. (Scale bar is 30 μm).

As shown in Figure 6, plate-like structures with lengths of 2 to 10 μm were formed owing to the inhomogeneity of the local chemical composition. However, most structures were lath-shaped martensite with a width of less than approximately 1 μm. At the time of EBSD measurement, the BCC and FCC structures were used for imaging. However, analysis showed that more than 99.6% was BCC, and about 0.4% was FCC. This means that when the specimen was cooled to 60 °C, all existing austenite underwent a phase transformation in martensite as Figure 6a [9–13].

The IPF photo in Figure 6b reveals martensite consisting of laths with a maximum number of 24 cases from one PAGB. This made it possible to track the grain boundary, which was the parent phase, immediately before water cooling [11].

### 3.4.4. Quantitative Analysis of Laths of the QT and AT Specimens

Table 5 compares the length and width of the laths analyzed using the IPF maps in Figure 5b (AT specimen) and Figure 6b (QT specimen). Here, the lath length of the AT specimen was measured with a maximum of 29.4 μm, a minimum of 7.1 μm, and an average

of 15.5 µm, and the width of the AT lath was measured with a maximum of 5.29 µm, a minimum of 0.59 µm, and an average of 1.76 µm.

**Table 5.** Comparison of length and width of laths in the AT and QT specimens.

| Dimension | AT | | QT | |
|---|---|---|---|---|
| | Length (µm) | Width (µm) | Length (µm) | Width (µm) |
| Average | 15.5 | 1.76 | 3.06 | 0.81 |
| Max. | 29.4 | 5.29 | 9.37 | 2.76 |
| Min. | 7.1 | 0.59 | 2.20 | 0.55 |

In addition, the lath length of the QT specimen was measured with a maximum of 9.37 µm, a minimum of 2.2 µm, and an average of 3.06 µm, and the width of the QT lath was measured with a maximum of 2.76 µm, a minimum of 0.55 µm, and an average of 0.81 µm.

Here, the length of the lath of the duplex AT specimen was approximately five times that of the single martensite QT specimen, and the width was about twice that of the QT specimen. The average length and width of the lath of the AT specimen were longer and wider than those of the QT specimen. The good lath properties of duplex bainite AT specimens were in good agreement with the tendency that mechanical properties such as tensile strength, hardness, elongation, and impact energy are better than those of the single martensite QT specimen, and it is judged as a good parameter to predict these properties such as the previous study [2].

### 3.5. Comparison of S-N Curves and VHCF Characterristics According to Microstructure

Figure 7 compares the S-N curves of the spring steels obtained by rotary bending fatigue test. The S-N data of the AR specimen at room temperature are shown as '∆' in Figure 7, and the fatigue limit is about 450 MPa. Number 2 on figure means that two data points have overlapped, and arrows in VHCF represent no failure in the current fatigue life.

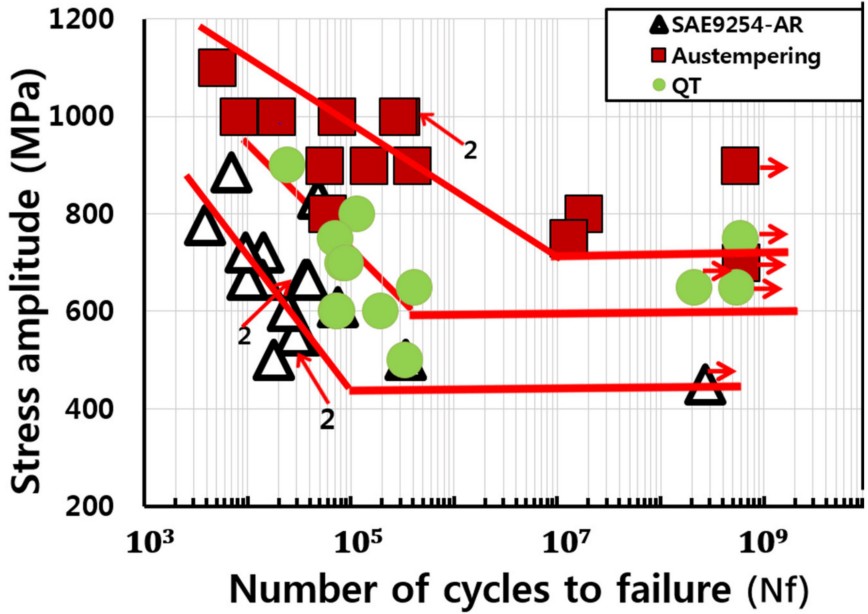

**Figure 7.** Comparison of S-N curves of spring steel before (AR) and after the heat treatment (AT and QT).

In addition, the martensite QT specimens are shown as '🟢' mark with fatigue limit of 600 MPa, which is about 33% increased value than that of AR specimen. Duplex bainite AT

specimens are shown as '■' mark with fatigue limit of 700 MPa, which is about 56% and 17% increased value over the AR and QT specimens, respectively.

The increase in fatigue strength of bainite and retained austenite AT specimens compared to single martensite QT specimens is thought to be due to changes in the shape of the bainite microstructure as long and wide laths and duplex microstructures (Figures 5 and 6). The structural transformation with bainite and retained austenite phase was analyzed for the excellent strength, ductility, and fatigue limit of spring steel specimens. These excellent increases in physical properties seem to be in good agreement with current and structural transformation studies on quantitative analysis of lath length and width.

Other studies showed that spring steels with duplex structures of bainite and martensite showed better strength and toughness than the single martensite structure [1–7]. A study about fatigue behavior of high strength steel with duplex phase of bainite and martensite by Wei et al. [13] reported that it showed higher fatigue strength and lower crack propagation rate.

### 3.6. Comparison of S-N Curves and UNSM Effects According to Microstructure

### 3.6.1. UNSM Treatment Effect of AR Specimen

Figure 8 is indicated by the 'Δ' mark in the S-N data obtained for the AR specimen, and the fatigue test results of the AR specimen treated with UNSM are shown as '▲' mark for comparison. Here, the UNSM-AR specimen showed fatigue limit at 600 MPa. This indicates that the fatigue limit increased about 33% compared to 450 MPa for the AR specimen. These UNSM surface treatment effects are similar to the results of other reports [29–33], and the UNSM-AR specimen showed a significantly increased fatigue life in the VHCF life region [15–25] compared to the AR specimen.

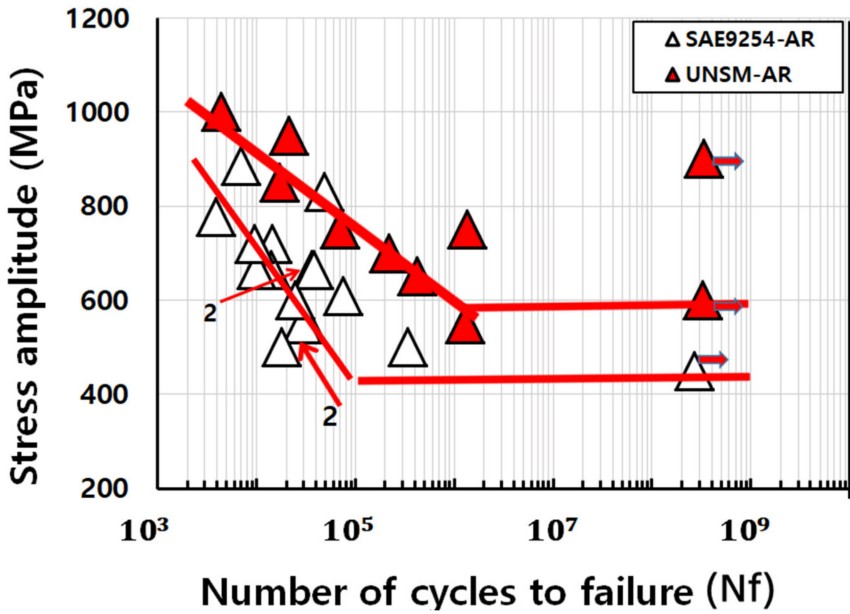

**Figure 8.** Comparison of S-N curves before and after UNSM treatment of the AR specimens.

### 3.6.2. UNSM Treatment Effect of AT Specimen

Figure 9 compares the S-N fatigue data of '□' mark on the duplex bainite AT specimen with '■' mark on the processed UNSM-AT specimen. UNSM-AT shows a fatigue limit at 1050 MPa, which is about 50% increase in fatigue limit from 700 MPa for AT specimen.

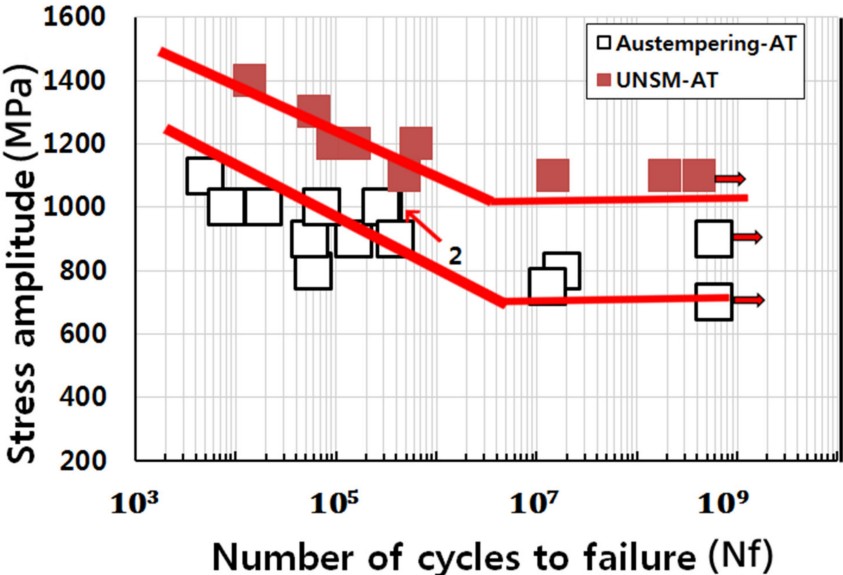

**Figure 9.** Comparison of S-N curves before and after UNSM treatment of the AT specimens.

It can be seen that the S-N fatigue data of the UNSM-AT specimen increase the fatigue limit by 50% greater than that of the AT specimen, and the scatter band of the data is small. This can be explained as an advantage of UNSM processing. That is, when UNSM treatment is performed, the surface is changed to being nanostructured, decreased in surface roughness, improved surface hardness, and large and deep compressive residual stress is formed on the surface.

### 3.6.3. UNSM Treatment Effect of QT Specimen

Figure 10 compares the fatigue data of '○' mark on the single martensite QT specimen with '●' mark on the processed UNSM-QT specimen. The fatigue limit of the UNSM-QT specimen is about 900 MPa, which is about a 50% increase in fatigue limit from 600 MPa of QT specimen.

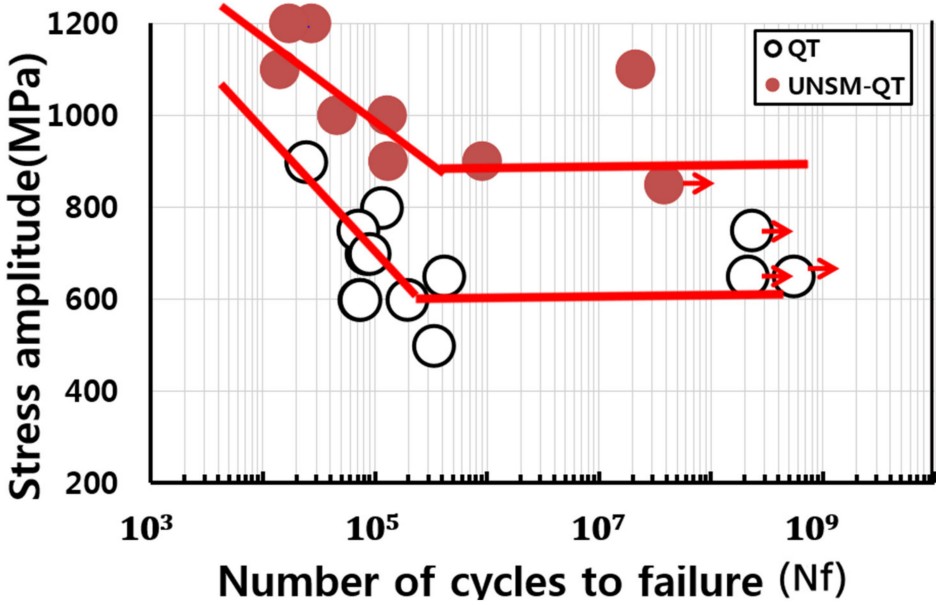

**Figure 10.** Comparison of S-N curves before and after UNSM treatment of the QT specimens.

In this study, UNSM-AR, UNSM-AT and UNSM-QT specimens treated with UNSM have about 33~50% better fatigue properties than untreated AR, AT, and QT specimens. This result is similar to other reports because the UNSM treatment resulted in a nanostructured surface, a decrease in surface roughness, improved surface hardness, and large and deep compressive residual stress on the surface [29–33].

*3.7. Observation of Fatigue Fractured Surfaces According to Microstructure*

3.7.1. AT Specimen

In Figure 11, the fish-eye crack, which is the part where the crack initiated, is indicated by arrows (center). In this specimen, the FGA (fine granular area) is indicated by a circle as an internal originating fracture type. This is a completely different result from the fish-eye cracking, initiation, growth, and coalescence process of common bearing steels [15–17].

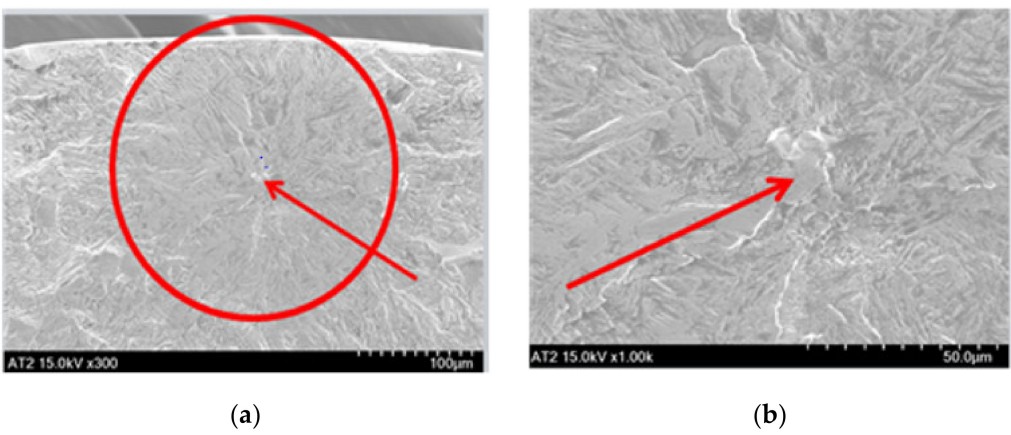

(**a**)                                                                                    (**b**)

**Figure 11.** SEM photographs of a fatigue fractured AT specimen (800 MPa, Nf = 1.8 × $10^7$) when observed at (**a**) 300× and (**b**) 1000×.

Figure 12 shows the SEM fracture surface of the AT specimen (1000 MPa, Nf = 3 × $10^5$) observed at 500 to 3000×. Here, the fisheye crack initiation site is indicated by a small square. This is an internal originating fracture type, circled FGA, and the results are different from other reports of fisheye crack initiation modes in common bearing steels [15–22].

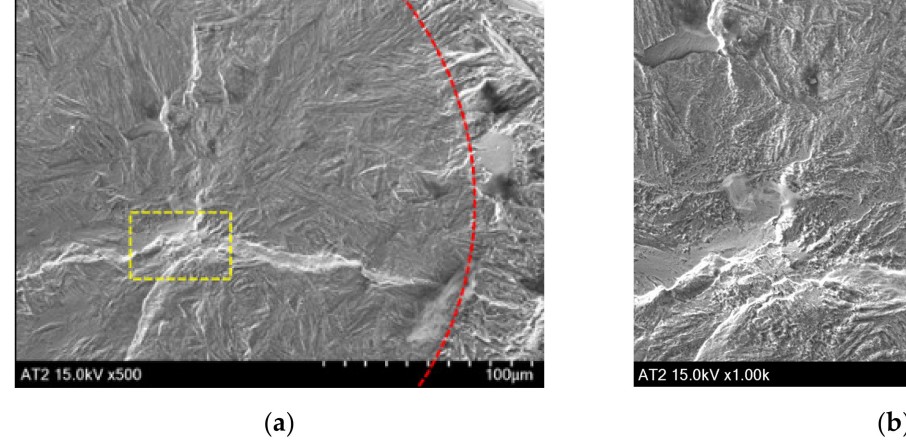

(**a**)                                                                                    (**b**)

**Figure 12.** *Cont.*

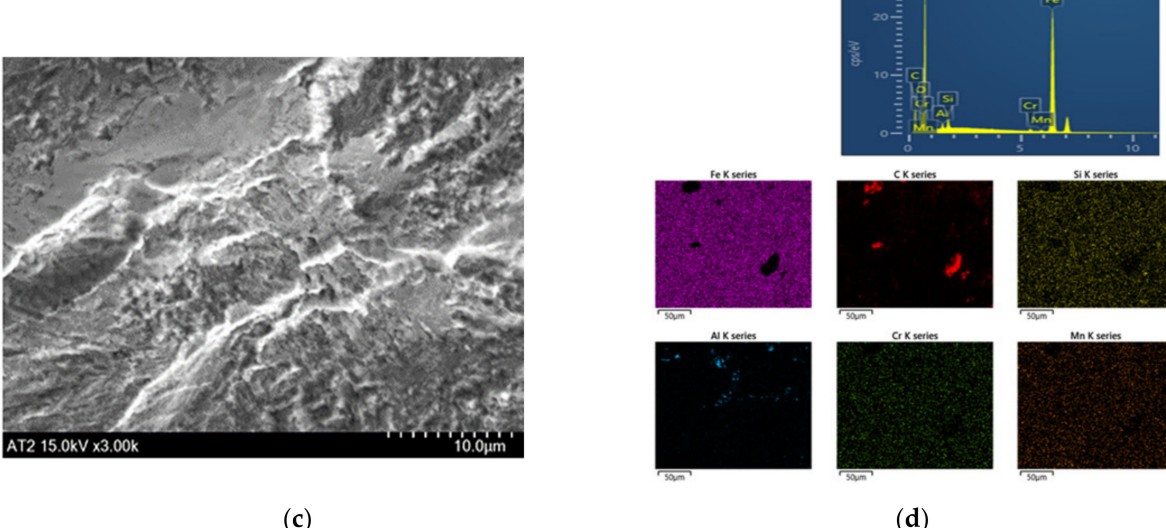

(c)  (d)

**Figure 12.** The fractography of a fisheye crack center on the AT specimen (1000 MPa, Nf = $3 \times 10^5$). (**a**) 500×, (**b**) 1000×, (**c**) 'facet' by 3000, and (**d**) data of EDS analysis of the crack initiation site in (**a**) indicated by a square.

Here, the traces of TiN and $Al_2O_3$ inclusions often observed in the center of fisheye cracks were not observed in structurally transformed to the duplex bainite AT specimen. The bainite lath observed with EBSD in Figure 5 can be easily seen in Figure 12b. The initiating site of the split type fisheye crack is indicated by an arrow in Figure 12c, and its width is a plane of about 35 μm.

The results of EDS analysis of the crack initiation site in Figure 12a, indicated by a square, are 11.51% C, 1.05% Si, 0.67% Cr, 0.61% Mn, and 0.57% Al, as shown in Figure 12d. This analysis result is slightly different from the main component of the specimen, but is the same as the fracture surface analysis results of other AT specimens. For reference, the chemical composition of the spring steel was 0.55% C, 1.5% Si, 0.7% Mn, and 0.7% Cr. Therefore, the fatigue crack initiated at a place where the carbon content was 21 times higher.

Fisheye cracks of bainite AT spring steel were similar to those formed in the 'facet' of Ti alloy, and occurred in a split type where there were no inclusions. It is also not a fractured fisheye crack that typically initiates and grows from an internal inclusion [15–22].

This is consistent with Nie's result of the fisheye cracks formed on the 'facets' in the spring steel UFT test [1]. In general, fisheye cracks are the preferred type of fracture in VHCF, but no fisheye cracks were found in QT specimens and AR specimens in VHCF.

### 3.7.2. QT Specimen

Figure 13a is SEM photographs of the fracture surface of single martensite QT specimen (650 MPa, Nf = $7.4 \times 10^6$). The arrow in Figure 13b indicates the location of fatigue crack initiation, and the semicircle in the figure indicates the morphology of crack propagation.

Figure 13d,e show EDS analysis data of the rectangular area of Figure 13c. In addition, Figure 13c shows the same place as the rectangular area of Figure 13a. In addition, the main component of this sample is shown in photos and data for each of the six components. As a result, the components of each element were uniformly distributed, and the contents of each element were analyzed as Si, 0.92%, Cr, 0.64%, and Mn, 0.59%, which were similar to the main component of the specimen except for C, 18.52%. Therefore, it seems that the central part of the test piece tends to have a high proportion of carbon.

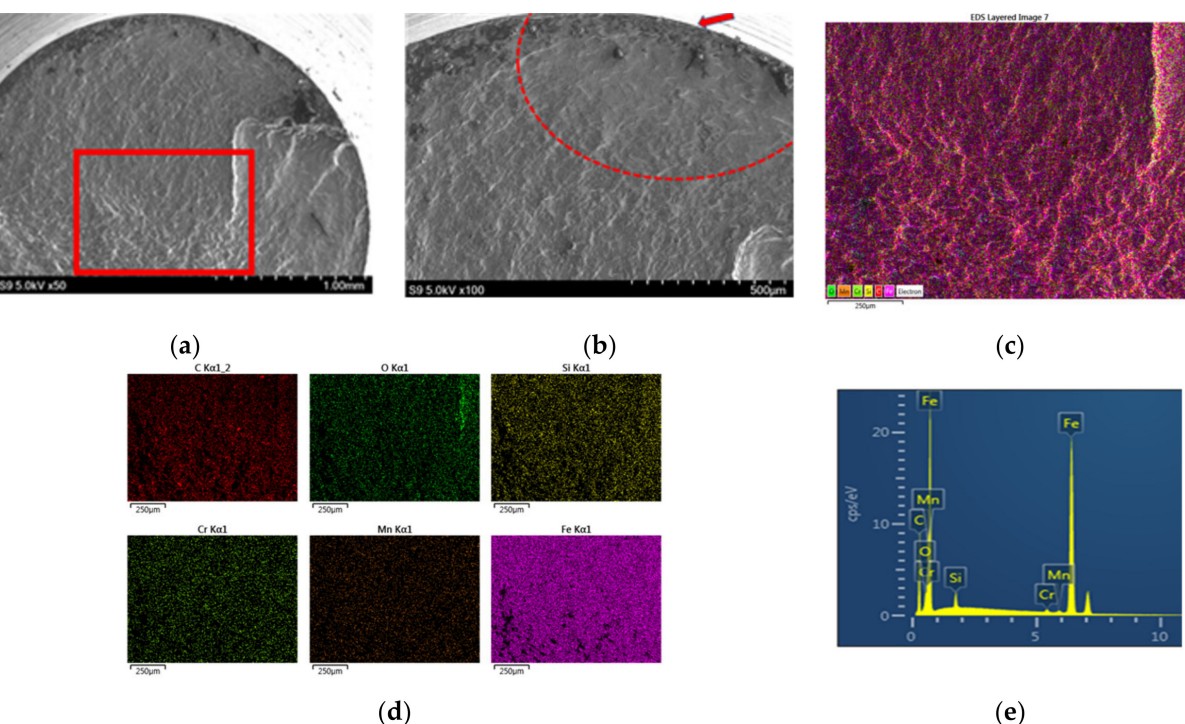

(a)    (b)    (c)

(d)    (e)

**Figure 13.** SEM photographs of the fracture surface of the QT specimen (650 MPa, Nf = 7.4 × 10$^6$). (**a**) 50×, (**b**) 100×, (**c**) 100×, (**d**,**e**) example of EDS analysis in (**c**).

### 3.7.3. UNSM-Treated AT Specimen

Figure 14 is SEM photographs of the fracture surface of UNSM-treated AT specimen (1100 MPa, Nf = 1.47 × 10$^7$). The fisheye crack, which is the crack initiation part, is circled in Figure 14b, and an arrow indicates that the central part of the crack initiating site is magnified and observed 100× to 2000×, respectively. The FGA is indicated by a circle in the fracture morphology that occurred inside the UNSM-treated fatigue specimen. In addition. traces of inclusions mainly composed of TiN and Al$_2$O$_3$, which are often observed in the center of fish-eye cracks of ordinary bearing steels, were not founded in this study.

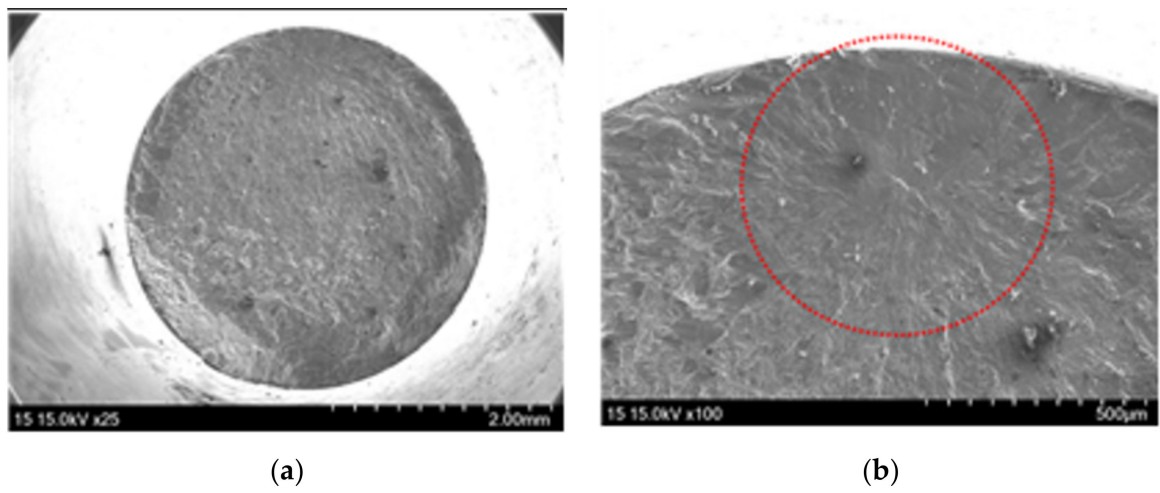

(a)    (b)

**Figure 14.** *Cont*.

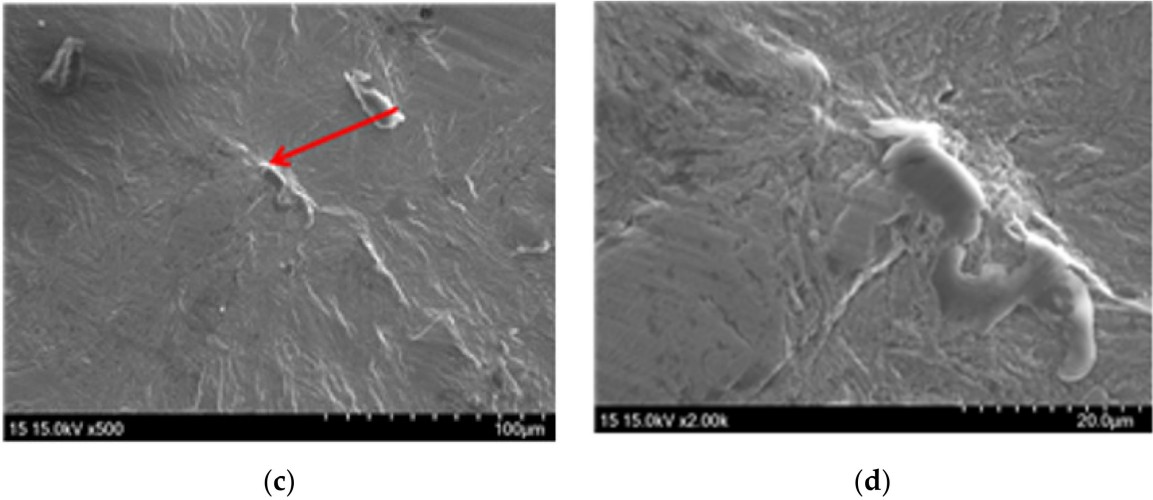

(c)                                         (d)

**Figure 14.** SEM photographs of the fracture surface of UNSM-treated AT specimen (1100 MPa, Nf = 1.47 × 10⁷). (**a**) 30×, (**b**) 100×, (**c**) 500×, and (**d**) 2000×.

*3.8. Tribology Characteristics According to Microstructure*

3.8.1. Variation in Friction Coefficient

Figure 15 shows the variation of friction coefficient using reciprocating tribology tester under 50 N and 100 N until 1800 s in dry sliding condition. Figure 15a compares the tribology test results of AT and QT specimen, shows a slight variation in friction coefficient, but it is kept at 0.6~0.94 under 50 N and at 0.78 under 100 N after about 450 s. Friction coefficients are more affected by normal load than specimens, and the result of 100 N shows less variation than that at 50 N.

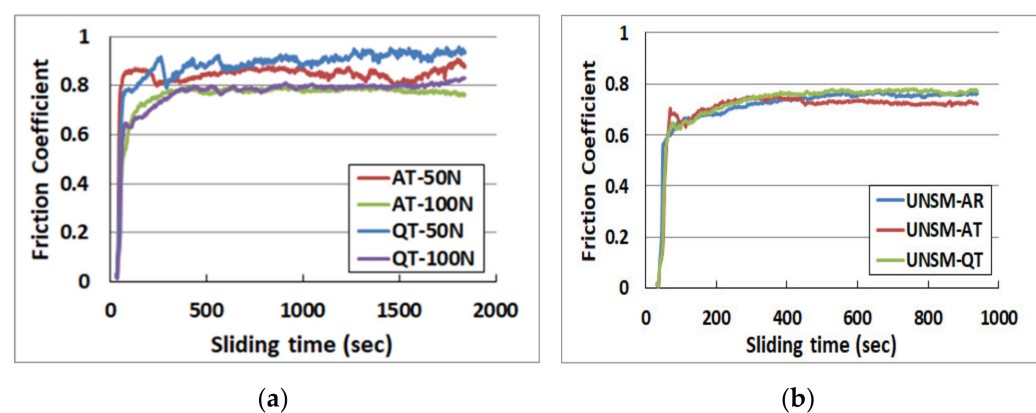

(a)                                         (b)

**Figure 15.** Variation in friction coefficient of AT, QT, and UNSM treated specimens under reciprocating dry sliding condition.

Figure 15b shows the experimental result of variation in friction coefficient as a function of time measured under 100 N until 1200 s in a dry sliding condition treated under the UNSM conditions of Table 1. Variation in the friction coefficient is kept at 0.73~0.78 in each specimen, and the UNSM treated specimen shows smaller and more stable values than the untreated specimen, as seen in Figure 15a. It was identified that the UNSM treated specimen showed better tribology characteristics in spring steel than some other steels, with better fatigue properties than the untreated specimen [29–33].

3.8.2. Comparison of Wear Amount and Tensile Strength According to Microstructure

Table 6 and Figure 16a compare the weight loss after the tribology test, and wear amount was increased at 100 N compared to 50 N. Weight loss of QT and AT specimen at

50 N and 100 N was large as much as 53~90% compared to AR specimen. The significant reduction in the amount of wear due to the morphology of the microstructure is judged to have good material properties in which the duplex bainite microstructure with large lath length and width has a lower wear amount than other microstructures.

**Table 6.** Comparison of weight loss according to microstructure.

| Specimen and Static Load | Weight Loss (g) | Percentage (%) | Specific Wear Rate [mm$^3$/(N·m)] |
|---|---|---|---|
| AR-50 N | 0.0029 | Base | $2.04 \times 10^{-4}$ |
| QT-50 N | 0.0011 | −62.1 | $0.78 \times 10^{-4}$ |
| AT-50 N | 0.0003 | −89.7 | $0.21 \times 10^{-4}$ |
| AR-100 N | 0.0038 | Base | $1.34 \times 10^{-4}$ |
| QT-100 N | 0.0018 | −52.6 | $0.64 \times 10^{-4}$ |
| AT-100 N | 0.0008 | −78.9 | $0.28 \times 10^{-4}$ |
| UNSM-AR-100 N | 0.0013 | Base | $0.69 \times 10^{-4}$ |
| UNSM-QT-100 N | 0.0008 | −38.5 | $0.42 \times 10^{-4}$ |
| UNSM-AT-100 N | 0.0002 | −84.6 | $0.11 \times 10^{-4}$ |

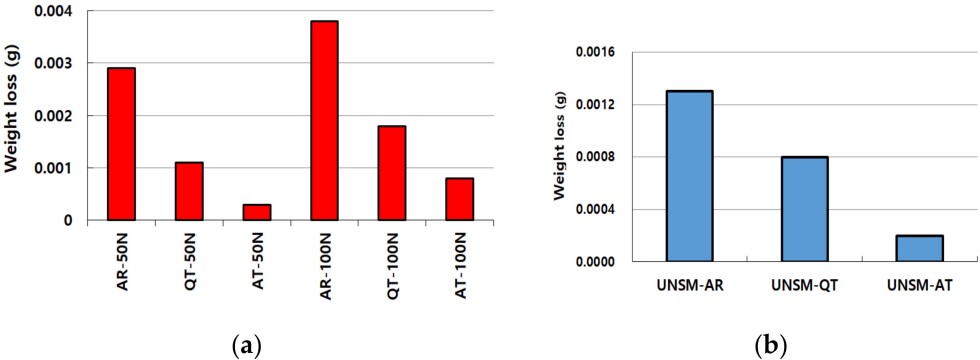

(**a**)                    (**b**)

**Figure 16.** Comparison of weight loss of the specimens (**a**) and the UNSM treated specimens (**b**) according to microstructure.

Figure 16b compares the weight loss of UNSM treated spring steel according to microstructure after tribology test under 100 N until 1200 s at dry sliding condition. The UNSM-QT and UNSM-AT specimens reduced the weight loss by approximately 38 to 85% compared to the UNSM-AR specimen, and the UNSM-AT specimen had the smallest weight loss. Improved wear resistance of UNSM-treated specimens compared to untreated specimens is due to formation of nano-surface structure, improved surface hardness, and large and deep compressive residual stress at the surface by UNSM treatment, as shown in other studies [29–33].

3.8.3. Comparison of Specific Wear Rate According to Microstructure

Specific wear rate (SWR) calculated from the measured weight loss is compared in Table 6. This specific wear rate is generally used as a parameter to compare the wear characteristics, and is given as Equation (1).

$$W_e = \frac{\Delta m}{\Delta t} \frac{1}{v \rho F_N} \tag{1}$$

Here, $W_e$ is specific wear rate [mm$^3$/(N·m)] to compare the degree of wear of materials. $\Delta m$ is amount of wear after test, $\Delta t$ is testing time, $v$ is test speed, $\rho$ is material density, and $F_N$ is normal load [6,26–28].

Figure 17 is a graph showing linear inverse relationship between the specific wear rate and the tensile strength of specimen. That is, the duplex bainite AT specimen with the

largest tensile strength showed the least specific wear rate. The specific wear rate of UNSM treated specimen at 100 N is also compared, shown as '♦' in this figure. Specific wear rate of UNSM treated specimen also showed a linear inverse relationship with tensile strength, three UNSM treated specimens with three microstructures showed the smallest specific wear rate, and the duplex bainite AT specimen with the largest tensile strength showed the smallest specific wear rate.

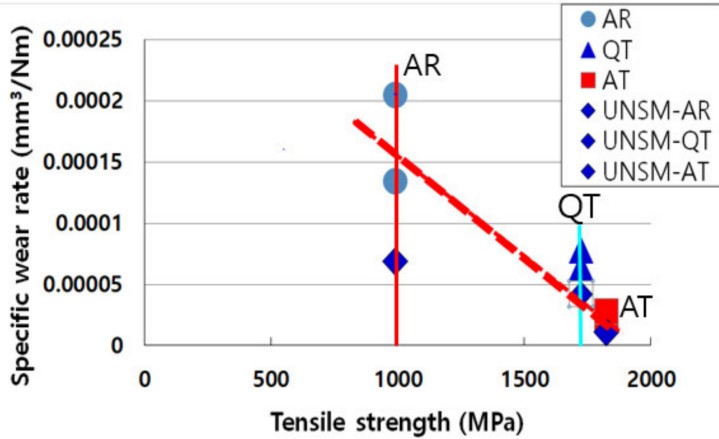

**Figure 17.** Correlation between specific wear rate and tensile strength.

3.8.4. SEM Observation of Wear Surface According to Microstructure

Figures 18 and 19 show SEM observations of the central part of the wear surfaces of AT and QT specimens after the wear test under 50 N until 1800 s in dry sliding conditions. The QT specimen, which had about 62% less wear than AR, had a rougher wear surface than the AT specimen, which had about 90% less wear. The UNSM-AT specimens and the UNSM-QT specimens show very similar images to those in Figures 18 and 19, respectively, except for the wear rate reduction.

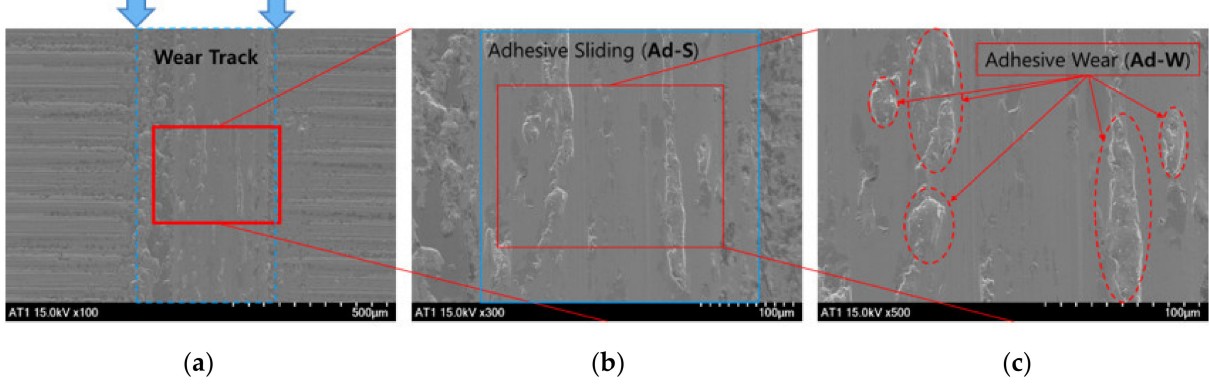

(a)  (b)  (c)

**Figure 18.** SEM images at the center of wear track of the AT specimen by (**a**) 100×, (**b**) 300×, and (**c**) 500×.

Figure 18 is an SEM image of duplex bainite AT specimen with two arrows in Figure 18a indicating the position of wear track and surface state. Figure 18b,c are SEM images magnified at the central part indicated by the rectangle of Figure 18a at 300× and 500×, respectively.

The AT specimen showed mainly smooth surface in the wear track and adhesive sliding (Ad-S) types were observed in the SEM, as seen in Figure 18b. However, in the enlarged Figure 18c, it is judged to be a local adhesive wear (Ad-W) type. That is, as indicated by several arrows, it can be observed that the localized surface structure reached fracture toughness and was peeled off by adhesion with other particles.

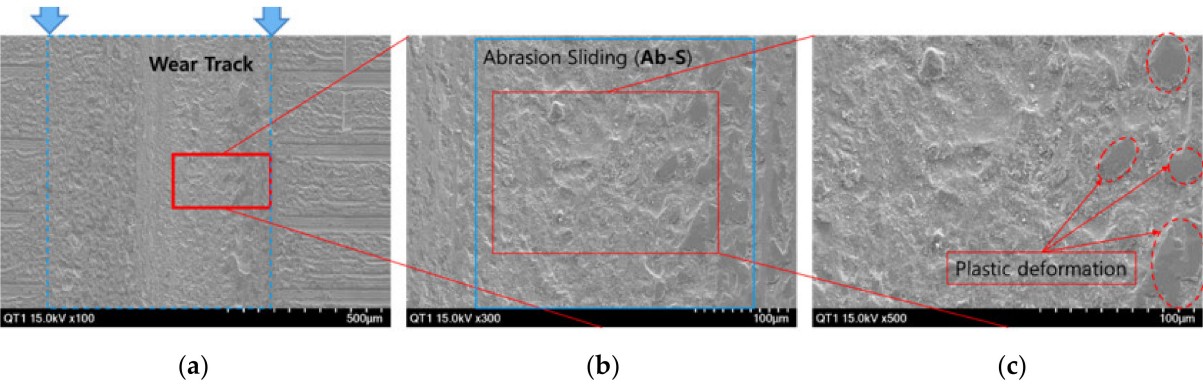

**Figure 19.** SEM images at the center of wear track of the QT specimen by (**a**) 100×, (**b**) 300×, and (**c**) 500×.

Figure 19 is an SEM image of a martensite QT specimen, with the two arrows in Figure 19a indicating the positions of the wear track and surface state. Figure 19b,c are enlarged SEM images magnified at the central part, indicated by the rectangle in Figure 19a at 300× and 500×, respectively.

SEM observation of Figure 19b of QT specimen showed mainly rough surface in the wear track and surface formation by abrasion sliding (Ab-S) in the sliding motion. Figure 19c shows some plastic deformation as indicated in the photograph, and rough surface by abrasion wear (Ab-W) which occurred by a combination of micro polishing, micro cutting, and particle dropout. These are thought to come from high hardness and high possibility of brittleness, and lowered toughness by quenching.

The duplex bainite AT specimen showed more adhesion type in the wear surface, and single martensite QT specimen showed more abrasion wear type. The difference of wear type is thought to be related to tendency of less wear amount of the AT specimen than the QT specimen.

## 4. Conclusions

The VHCF, HCF, tribological properties, and UNSM effects of spring steels with duplex bainitic and single martensitic microstructures were quantitatively studied through fracture mechanics and fracture surface analysis methods from the engineering and industrial point of view to improve durability and weight reduction in spring steels. The results are as follow:

(1) The bainite and retained austenite AT specimens exhibited a fatigue limit of 700 MPa. The martensite QT specimen showed a fatigue limit at 600 MPa. This is a 56% and 33% increase in fatigue limit for AR specimens compared to 450 MPa, respectively. In particular, the increase in fatigue strength of AT specimens compared to QT specimens in the VHCF range is believed to be due to changes in microstructural characteristics by bainite and retained austenite microstructure.

(2) Fisheye cracks in duplex bainite AT specimens are not the type of fatigue cracks originating from internal inclusions, but are similar to 'facet internal cracks' that initiated in the absence of inclusions. Generally, fisheye crack fracture mode is preferred in VHCF, but fisheye crack was not found in the QT and the AR specimens at all.

(3) The UNSM-treated-AT, -QT, and -AR specimens showed fatigue limits that were about 50~33% higher than the untreated AT, QT, and AR specimens. Fisheye cracks were not detected in the fracture surfaces of UNSM-QT and UNSM-AR specimens, but were only detected in a VHCF range in UNSM-AT specimens.

(4) The weight loss for QT and AT specimens was significantly reduced by approximately 53% to 90% compared to that for AR specimen. In addition, the weight loss for UNSM treated QT and -AT specimens was reduced by about 38% to 85% compared with

that of the UNSM treated AR specimens. The number of UNSM-AT specimens was reduced the least.

(5) The specific wear rates of the AT and QT and the UNSM treated specimens were linearly inversely proportional to the tensile strength. In addition, the three UNSM treated specimens had the lowest specific wear rate.

(6) The duplex bainitic AT specimen had a lot of adhesive sliding (Ad-S) areas in the wear surface, but locally developed adhesive wear (Ad-W). In the case of the single martensitic QT specimen, local plastic deformation occurred, and many traces of abrasion sliding (Ab-S) were observed. The difference of wear type is thought to be related to the tendency to lesser wear in the AT specimen than the QT specimen.

**Author Contributions:** Data curation, M.S.S.; Formal analysis, S.H.N.; Funding acquisition, M.S.S. and S.H.N.; Investigation, C.M.S. and Y.S.P.; Methodology, C.M.S. and Y.S.P.; Project administration, S.H.N.; Writing—original draft preparation, C.M.S.; Writing—review and editing, C.M.S., M.S.S., S.H.N. and Y.S.P. All authors have read and agreed to the published version of the manuscript.

**Funding:** This research was supported by the Development of Reliability Technology of Standard Measurement for Hydrogen Convergence Station funded by the Korea Research Institute of Standards and Science (KRISS-2021-GP2021-0007). This work was supported by the National Research Foundation of Korea (NRF 2017033524) and the National Research Council of Science & Technology (NST CAP20032-200) grant funded by the Korean government (MSIT).

**Data Availability Statement:** Not applicable.

**Conflicts of Interest:** The authors declare no conflict of interest.

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
