# Peer review of "VHCF, Tribology Characteristics and UNSM Effects of Bainite and Martensite Spring Steels"

_metals, doi:10.3390/met12060901_

Round 1

Reviewer 1 Report

Due to the locality of fatigue and wear processes, the role of structural factors in shaping the level of fatigue life of bainite and martensite spring steels is extremely important. Therefore, the article is important for science and practice. However, there are a few suggestions to improve it:

1….43. An analysis of the literature in the range [1-14] seems to me to be very superficial, uninformative and non-specific. I propose to make it more understandable to readers of the article.

  1. In fig. 1 shows photographs of standard testing machines. In my opinion, it is better to show test schemes, image parameters, etc. in this figure. Now the size of this figure does not allow it to be deeply analyzed. And if you make the size of the picture convenient for analysis, it will be too big. Please, consider my recommendation.
  2. According to what standards were the fatigue test conditions chosen? It is necessary to justify the choice of frequency, stress ratio, etc.
  3. Scale of the yellow arrow and greek letter in fig. 3 do not look very aesthetically pleasing. I invite the authors to look at the article: https://www.sciencedirect.com/science/article/abs/pii/S1644966516300048. It also has arrows on SEM images, but they look more neat.
  4. Scale of points in fig. 8-10 does not allow normal analysis of the figure. The size of the dots must be reduced by 3 times.
  5. Micromechanisms of fracture of steels are shown in fig. 11-13. But are very sparsely described. Descriptions of the micromechanisms of fracture are not systematized. It needs to be supplemented. Please read the article recommended in 4.
  6. Linear relationships shown in fig. 17 are very uninformative because:

7.1. There are very few points, which does not allow one to see the nonlinearity of this dependence.

7.2. In nature, there are very few linear relationships between parameters, this is well known. In order to show a linear relationship between the parameters it is necessary to show and prove its physical and mechanical correctness.

This is not here, so I think that this figure should be removed from the article, as unscientific and unproven.

  1. The authors analyzed fatigue micromechanisms and wear micromechanisms. But their systematization is missing. I propose to add a table and systematize it, following the example of the article recommended in 4.

Author Response

Thank you so much for your careful comments and suggestions. Authors team have reviewed them modified as attached. 

Reviewer 2 Report

The paper reported a comprehensive study on tensile, fatigue and wear properties of a spring steel in the AS, QT and AT states. It contains useful information to engineers in the subject areas. Some minor points in presentation should be improved (or maybe just because the eye sight of the reviewer). 

  1. In discussion about Fig. 5 (perhaps Fig. 6, too), the authors said: "It can be seen that they have similar colors 218 locally, but it can also be confirmed that they don't just appear in the same color." Please clarify what are "they" by color?
  2. Pictures in Fig. 6 looks to be the same as in Fig. 5, but they are one for QT and one for AT microstructure. Please explain.
  3. About Fig. 14, it is said: "The FGA is indicated by a circle...", but the "circle" is missing.
  4. Page 14, line 395-397, "In addition, no fisheye cracks were found in the fracture surface observation results of the fatigue test of UNSM-treated UNSM-QT specimen and UNSM-AR specimen, and some of them occurred in VHCF in UNSM-AT specimen. Can the authors confirm crack nucleation occurred internally but just no fisheye feature in these samples or what (e.g., surface crack nucleation)?
  5. Page 15, line 438, FN, N goes into subscript.
  6. The authors showed the worn surfaces of QT and AT materials. For completeness, should also show that of UNSM and the discuss the wear mechanism on that too.

Author Response

Thank you so much for your careful comments and suggestions. Authors team have reviewed them and modified as attached, 

Reviewer 3 Report

Dear Authors,

The article submitted to me for review, entitled "VHCF, Tribology Characteristics and UNSM Effects of Bainite and Martensite Spring Steels" deals with the study of microstructure, mechanical properties, and tribological properties of SAE9254 spring steel. The Authors describe the research in a substantive correct manner, analyse the obtained results and draw the right conclusions. The content of the article has a high practical industrial importance, especially due to the increasing attention given to the environmental and social sustainability issues of production processes. I rate the reviewed article very positively in terms of the scope of research and the essence of the information provided. Authors must checks on terminology, uniformity of language, spelling etc. I congratulate the Authors of the high level of research and to encourage further work undertaken on the subject. I accept the article for publication without corrections.

Round 2

Reviewer 1 Report

  1. Unfortunately, the authors did not take into account my remark. Forced to repeat it:

- references [1–14] and [15–28] are, in my opinion, incorrect from a scientific point of view. They do not allow us to understand how the results of the author differ from the given papers [2], [3], [4], [5], [6], [7], etc.

- in my opinion, the author should have analyzed these publications, for example: in [2] it was shown ..., in [3] was obtained ..., etc. After all, this is precisely the main value of the Introduction of the article. Unfortunately, the authors ignore this traditional approach and simply make mechanical references, combining dozens of articles into them, without the slightest analysis of their content and results. I consider such references unacceptable.

  1. It is necessary in section 2 to specify the dimensions of the samples.

  1. It is necessary to indicate the test standards and provide references to it in the references list, this is necessary in order to:

- show that the tests are in accordance with generally accepted standards;

- show that the results are correct;

- substantiate test modes. Now they are not justified. And if there are doubts about the methods, then the results are considered doubtful;

- to make it possible for the readers of the article to repeat this experimenter.

The choice of test parameters (frequency and cycle stress ratio) must be justified.

  1. The inscriptions on the figures must be neat. Why do we need a "supercapital letter" in fig. 3a? Perhaps it can be corrected for a more accurate one?

  1. Scale of points in fig. 8-10 does not allow normal analysis of the figure. The size of the dots must be reduced by 3 times. I have not seen such large "dots" in the figures in one article over the past 20 years.

  1. Micromechanisms of fracture of steels are shown in fig. 11-13. But are very sparsely described. Descriptions of the micromechanisms of fracture are not systematized. It needs to be supplemented. From the point of view of these figures, the authors did not substantiate the fundamental fractographic patterns that establish the relationship between the endurance limit and the extent of defects or other characteristics of the material. I don't see it.

  1. Unfortunately, I do not agree with the opinion of the authors about figure 17. Any correlation must have a physical and mechanical basis. If we look for correlations between events that are not physically connected, then this is not a science, but magic :)

The results of fig. 17 I consider incorrect and physically strange.

Linear relationships shown in fig. 17 are very uninformative because:

7.1. There are very few points, which does not allow one to see the nonlinearity of this dependence. 7.2. In nature, there are very few linear relationships between parameters, this is well known. In order to show a linear relationship between the parameters it is necessary to show and prove its physical and mechanical correctness. This is not here, so I think that this figure should be removed from the article, as unscientific and unproven.

  1. Fundamental analytical relationships that establish the connection between the endurance limit and other characteristics of the material and test parameters are not shown, based on the data in Fig. 8.9. A generalized equation has not been proposed that describes the dependence of the endurance limit of the studied materials on the microstructural components of the material using a single curve. It must be invariant with respect to the dimensions of the samples.

9. The authors analysed fatigue micromechanisms and wear micromechanisms. But their systematization is missing. I propose to add a table and systematize it in table, or in scheme.